# Intranasally Administered Exosomes from Umbilical Cord Stem Cells Have Preventive Neuroprotective Effects and Contribute to Functional Recovery after Perinatal Brain Injury

**DOI:** 10.3390/cells8080855

**Published:** 2019-08-08

**Authors:** Gierin Thomi, Marianne Joerger-Messerli, Valérie Haesler, Lukas Muri, Daniel Surbek, Andreina Schoeberlein

**Affiliations:** 1Department of Obstetrics and Feto-Maternal Medicine, Inselspital, Bern University Hospital, University of Bern, 3010 Bern, Switzerland; 2Department for BioMedical Research (DBMR), University of Bern, 3012 Bern, Switzerland; 3Graduate School for Cellular and Biomedical Sciences, University of Bern, 3012 Bern, Switzerland; 4Neuroinfection Laboratory, Institute for Infectious Diseases, University of Bern, 3012 Bern, Switzerland

**Keywords:** perinatal brain injury, white matter injury, gray matter injury, hypoxia-ischemia, mesenchymal stem cells, umbilical cord, exosomes, intranasal, neuroregeneration, memory

## Abstract

Perinatal brain injury (PBI) in preterm birth is associated with substantial injury and dysmaturation of white and gray matter, and can lead to severe neurodevelopmental deficits. Mesenchymal stromal cells (MSC) have been suggested to have neuroprotective effects in perinatal brain injury, in part through the release of extracellular vesicles like exosomes. We aimed to evaluate the neuroprotective effects of intranasally administered MSC-derived exosomes and their potential to improve neurodevelopmental outcome after PBI. Exosomes were isolated from human Wharton’s jelly MSC supernatant using ultracentrifugation. Two days old Wistar rat pups were subjected to PBI by a combination of inflammation and hypoxia-ischemia. Exosomes were intranasally administered after the induction of inflammation and prior to ischemia, which was followed by hypoxia. Infrared-labeled exosomes were intranasally administered to track their distribution with a LI-COR scanner. Acute oligodendrocyte- and neuron-specific cell death was analyzed 24 h after injury in animals with or without MSC exosome application using terminal deoxynucleotidyl transferase dUTP nick end labeling (TUNEL) assay and immunohistochemical counterstaining. Myelination, mature oligodendroglial and neuronal cell counts were assessed on postnatal day 11 using immunohistochemistry, Western blot or RT-PCR. Morris water maze assay was used to evaluate the effect of MSC exosomes on long-term neurodevelopmental outcome 4 weeks after injury. We found that intranasally administered exosomes reached the frontal part of the brain within 30 min after administration and distributed throughout the whole brain after 3 h. While PBI was not associated with oligodendrocyte-specific cell death, it induced significant neuron-specific cell death which was substantially reduced upon MSC exosome application prior to ischemia. MSC exosomes rescued normal myelination, mature oligodendroglial and neuronal cell counts which were impaired after PBI. Finally, the application of MSC exosomes significantly improved learning ability in animals with PBI. In conclusion, MSC exosomes represent a novel prevention strategy with substantial clinical potential as they can be administered intranasally, prevent gray and white matter alterations and improve long-term neurodevelopmental outcome after PBI.

## 1. Introduction

Perinatal brain injury (PBI) is a major complication in infants born prematurely causing substantial mortality and neurodevelopmental impairment. Long-term neurodevelopmental sequelae in survivors of PBI include behavioral, cognitive, motor and psychological problems [1] as well as memory deficits [2] which negatively influence schooling performance and quality of life of affected children and may even persist into adulthood [3,4,5,6,7].

The pathogenesis of perinatal brain injury is complex but is thought to involve both fetal inflammation and neonatal hypoxia/ischemia causing the activation of brain resident immune cells, excitotoxicity and the generation of free radicals [8]. This subsequently leads to substantial white matter injury, as the preterm white matter is mainly populated by immature oligodendrocytes especially sensitive to hypoxic-ischemic and inflammatory insults [8]. Hence, PBI is characterized by white matter injury within the periventricular white matter and corpus callosum, deficits in myelination and decreased white matter volumes detected by MRI [9]. Neuropathological studies of postmortem brains from preterm infants indicate that myelination deficits in PBI are mediated by immature oligodendrocyte-specific cell death, as immature oligodendrocytes seem to be particularly sensitive to hypoxic-ischemic and inflammatory insults [10,11]. In contrast, other studies found no evidence of oligodendrocyte loss but identified impaired differentiation of vulnerable immature oligodendrocytes as the underlying cause for alterations in myelination [12,13].

Additionally, PBI is also characterized with substantial gray matter injury within the cortex, thalamus and hippocampus [14]. Neuropathological studies of preterm brains with PBI revealed primary gray matter injuries such as apoptosis of developing axons [15] and gamma-aminobutyric acid (GABA)-expressing neurons [11] as well as the loss of hippocampal [16] and subplate neurons [17,18]. Other opposing studies showed only marginal neuronal loss but found evidence of neuronal maturation deficits such as immature dendritic arborization and altered synaptic signaling of neurons [19].

So far, only limited treatment options are available to confer neuroprotective or—regenerative effects in PBI. Promising therapeutic strategies for PBI should focus on preventing white and gray matter loss as well as on promoting oligodendrocyte and neuronal maturation and regeneration [14].

Mesenchymal stromal cells (MSC) represent a promising therapeutic strategy for neuroprotection and regeneration of both gray and white matter in PBI [20]. MSC have been shown to promote oligodendrocyte and neuronal proliferation [21] and to prevent neuronal cell death [22] in neonatal rats with hypoxic-ischemic brain injury. MSC have also been shown to bolster oligodendrocyte regeneration by boosting their lineage maturation in neonatal rats after combined inflammatory and hypoxic-ischemic brain injury [23]. MSC further induced neuronal and oligodendrocyte regeneration and thereby improved functional outcome in neonatal mice after hypoxic-ischemic injury [21,24]. The neuroregenerative effects of MSC were initially thought to occur via replacement of lost or injured cells. In recent years, however, increasing evidence suggests that MSC exert their therapeutic effects in CNS injuries rather via paracrine mechanisms [25], including the release of extracellular vesicles [26,27,28].

Extracellular vesicles (EV) such as microvesicles and exosomes mediate intercellular communication by transferring cargo from their cell of origin to their target cell [29]. While microvesicles (50 nm–1 µm in diameter) are formed and released by the budding of the plasma membrane, exosomes are small membrane vesicles (30–150 nm in diameter) derived from the endosomal compartment and released during the fusion of multivesicular bodies with the plasma membrane [29]. MSC-EV hold great therapeutic potential in PBI as they exhibit similar neuroprotective and -regenerative potential as their mother cells [30]. For example, MSC-derived extracellular vesicles protected against hypoxia-ischemia-induced myelination alterations and improved brain function in fetal sheep [31]. Further, MSC-derived microvesicles protected against gray and white matter apoptosis in a rat model of inflammation-induced preterm brain injury [26] while MSC-derived exosomes had neuroprotective and neuroregenerative effects on neuronal cells in an in vitro model of hypoxia-ischemia [32]. MSC-EV also resolve several safety issues associated with the transplantation of MSC such as the tumorigenic potential of proliferating cells, the host’s immunocompatibility and the risk of emboli formation, while their easy handling and storage allow for an off-the-shelf therapy. Hence we aimed to assess the neuroprotective effects of MSC-derived exosomes and their potential to improve behavioral outcome after PBI.

## 2. Materials and Methods

### 2.1. Establishment of Human Wharton’s Jelly-Derived Mesenchymal Stromal Cell (hWJ-MSC) Exosomes

Umbilical cords from healthy term deliveries by caesarean section (mean gestational age 38 ± 1.4 weeks, *n* = 7) were collected. The study was approved by the Ethics Committee of the Canton of Bern (reference numbers: KEK BE 090_07 and KEK BE 178_03), and all patients donating umbilical cords gave written informed consent. hWJ-MSC were isolated from the connective tissue of the umbilical cord (Wharton’s jelly) via enzymatic digestion as previously described [33]. hWJ-MSC were cultured in expansion medium consisting of Dulbecco’s modified Eagle’s medium (DMEM)/F12 supplemented with 10% fetal bovine serum (FBS), 2 mmol/L GlutaMAX™, 100 units/mL penicillin and 100 μg/mL streptomycin (Thermo Fisher Scientific, Waltham, MA, USA). At passage 4–6, cells were washed twice with PBS and the expansion medium was exchanged for serum-free medium containing DMEM/F12, 2 mmol/L GlutaMAX™, 100 units/mL penicillin and 100 μg/mL streptomycin only. After 36 h of incubation, hWJ-MSC culture supernatants were collected. Exosomes were isolated from the supernatant using serial centrifugations according to the protocol of Théry et al. [34] and as we had previously described [32]. The pelleted exosomes were resuspended in PBS, characterized as previously described [35] and stored at −20 °C.

### 2.2. Animal Model of Perinatal Brain Injury

A neonatal rat model of PBI was established using a combination of a hypoxic-ischemic and an inflammatory insult (Figure 1A). All animal procedures were approved by the Veterinary Department of the Canton of Bern, Switzerland (reference number: BE117/16; 28384). Wistar rat pups (Janvier Labs, Le Genest-Saint-Isle, France) on P2 were randomly assigned to four experimental groups independent of sexes: Healthy, PBI, PBI+exosomes (Exo), and PBI+IRDye^®^ 800CW-labeled Exo (800CW-Exo) (Figure 1B). Healthy control animals received a saline injection, were sham-operated, kept under normoxic conditions and received no exosomes. PBI and PBI+(800CW-)Exo animals received an intraperitoneal injection of 0.1 mg/kg LPS in saline (*Escherichia coli* 0111:B4; Sigma-Aldrich St. Louis, MO, USA), followed by the cauterization of the left common carotid artery 2 h later and exposure to hypoxia (8% O_2_/92% N_2_, 3 L/min) for 55 min (Figure 1A), as previously described [23]. Between the LPS injection and the ligation, PBI+Exo animals received exosomes in PBS (50 mg/kg) intranasally (i.n.), PBI+800CW-Exo animals received IRDye^®^ 800CW-labeled exosomes in PBS (10 mg/kg) i.n. and PBI animals received PBS i.n. only (Figure 1B). Exosomes were stained with IRDye^®^ 800CW according to the manufacturer’s instructions (LI-COR Biosciences, Lincoln, NE, USA). Drops of hyaluronidase (100 U in PBS/nostril, Sigma-Aldrich) were given 30 min before (800CW-) exosome or PBS administration to ensure an increased permeability of the nasal mucosa. Spontaneous mortality and weight gain during the experiment were documented.

### 2.3. Exosome Uptake Evaluation

For detecting the intranasally administered exosomes in the brain, PBI (*n* = 2) and PBI+800CW-Exo (*n* = 6) animals were sacrificed 30 min or 3 h post exosome administration on P2 by decapitation (Figure 1A). Brains were harvested and scanned with an Odyssey imaging system (LI-COR Biosciences, Lincoln, NE, USA). Image analysis was performed using Image Studio (LI-COR Biosciences) Version 3.1.

### 2.4. RNA and Protein Isolation

Healthy (*n* = 3), PBI (*n* = 6) and PBI + Exo (*n* = 5) animals were sacrificed on P11 with terminal sodium pentothal anesthesia and transcardial perfusion with PBS (Figure 1A). The QIAshredder and the All-prep DNA/RNA/Protein Mini Kit were used according to the manufacturer’s protocol (Qiagen, Hilden, Germany) to isolate RNA and protein. Total protein concentration was determined using the Bicinchoninic Acid Protein Assay Kit (Sigma-Aldrich). RNA concentration was measured using a NanoVue Plus™ spectrophotometer (Biochrom, Holliston, MA, USA). RNA purity was assessed by measuring the 260 nm/280 nm ratio and a ratio between 1.8 and 2.1 was considered as pure and high-quality RNA. Up to 2 μg RNA was reverse transcribed using the SuperScript III First-Strand Synthesis System (Thermo Fisher Scientific).

### 2.5. Gene Quantification by Real-Time Polymerase Chain Reaction (RT-PCR)

Gene expression was quantified using real-time reverse transcription polymerase chain reaction (RT-PCR). Gene expression of oligodendroglial myelin basic protein (Mbp) and neuronal microtubule-associated protein 2 (Map2) was quantified by real-time RT-PCR using the Taqman primer and probes gene expression assays for Mbp, ID Rn01399619_m1 and Map2, ID Rn0056046_m1. The PCR cycling program was run for 2 min at 50 °C, then for 10 min at 95 °C, followed by 45 cycles of 15 s at 95 °C and 1 min at 60 °C on a QuantStudio™ 7 Flex Real-Time PCR System (Thermo Fisher Scientific). The housekeeping gene glyceraldehyde-3-phosphate dehydrogenase was used as endogenous control and primer and probe sequences were adopted from RTPrimerDB [36]. Data were analyzed using the QuantStudio™ Real-Time PCR software (Thermo Fisher Scientific). Gene expression was calculated using the 2^−ΔΔCt^ method relative to total rat brain RNA (Amsbio, Abingdon, UK).

### 2.6. Western Blot Analysis

Proteins were separated by sodium dodecyl sulfate-polyacrylamide gel electrophoresis (SDS-PAGE) on a 4% to 20% gradient gel (Bio-Rad Laboratories, Inc., Hercules, CA, USA) and transferred onto polyvinylidene fluoride or nitrocellulose membranes (Thermo Fisher Scientific). Membranes were blocked either with 5% bovine serum albumin (BSA, Sigma-Aldrich) or 5% milk in Tris-buffered saline (TBS). Proteins were analyzed with rabbit antibodies against myelin basic protein (Mbp, 1:400, AB980, Thermo Fisher Scientific), microtubule-associated protein 2 (Map2, 1:500, ab24640, Abcam, Cambridge, UK) and β-Tubulin (1:4000, ab6046, Abcam) in 5% milk (for Mbp) or 5% BSA (for Map2 and β-Tubulin) in TBS-Tween 20 (TBS-T, Sigma-Aldrich) overnight at 4 °C. Horseradish peroxidase-coupled donkey anti-rabbit (1:1000, GE Healthcare Life Science, Piscataway, NJ, USA) antibody was used as a secondary antibody and incubated with the membranes for 1 h at room temperature. Binding was detected using the chemiluminescent Amersham ECL Prime Western Blotting Detection Reagent (GE Healthcare Life Sciences) on a Chemidoc XRS+ system (Bio-Rad). Pixel summation of individual bands was performed with ImageJ Software (NIH, Bethesda, MD, USA). The ratio between Mbp/Map2 and β-Tubulin was calculated for each animal.

### 2.7. Immunohistochemistry

For immunohistochemistry analysis, animals were sacrificed with terminal sodium pentothal anesthesia and transcardially perfused with PBS followed by 4% paraformaldehyde in PBS (Figure 1A). Brains were fixed in 4% paraformaldehyde for 24 h and embedded in paraffin. Coronal brain paraffin sections (6 µm) were cut, deparaffinized and rehydrated prior to antigen retrieval by heating sections to 100 °C in 0.1 M sodium citrate buffer for 12 min. Sections were blocked with 10% goat serum, 1% BSA in TBS-Tween 20^®^ (Sigma Aldrich) for 2 h at room temperature. For analysis of the corpus callosum and external capsule regions, coronal sections at Bregma 0.60 mm (P3) or −0.05 mm (P11) were used, respectively. Coronal sections at Bregma −1.40 mm from P3 brains were taken to analyze the hippocampal formation and the posterior parietal cortex [37]. For immunohistochemistry analysis, every third brain slice was selected in each region of interest and all slides were anonymized and assessed in a blinded way.

Cell death quantification was assessed in sections from Healthy (*n* = 4), PBI (*n* = 3) and PBI+Exo (*n* = 3) animals sacrificed on P3. Blocked sections were stained with a terminal deoxynucleotidyl transferase dUTP nick end labeling (TUNEL) In situ Cell Death Detection Kit according to the manufacturer’s protocol for labeling difficult tissues (chapter 3.3.4, version 17, Sigma-Aldrich). After the TUNEL staining, sections were incubated with primary antibodies for the neuronal marker neuronal nuclear antigen (NeuN) and the oligodendroglial marker oligodendrocyte transcription factor 2 (Olig2) (rabbit-anti-NeuN, ab177487, 1:500 and rabbit-anti Olig2, ab109186; 1:100, both Abcam) diluted in 10% goat serum, 1% BSA in TBS-Tween 20^®^ overnight at 4 °C. Finally, sections were incubated with Alexa fluor^®^ 594-conjugated secondary antibodies (Thermo Fisher Scientific) for 1 h at room temperature followed by counterstaining with 4′,6-diamidino-2-phenylindole (DAPI).

The count of mature oligodendrocytes was assessed in sections from Healthy (*n* = 4), PBI (*n* = 4) and PBI+Exo (*n* = 4) animals sacrificed on P11. For Ki67^+^Olig2^+^ and CNPase^+^Olig2^+^ stainings, blocked sections were incubated with primary antibodies against CNPase (1:100, c5922 Sigma-Aldrich), Ki67 (1:100, BD550609, BD Biosciences, San Jose, CA, USA) and Olig2 (1:200, ab109186, Abcam) in TBS-Tween 20^®^ overnight at 4 °C followed by secondary Alexa fluor^®^ 488-conjugated anti-mouse and Alexa fluor^®^ 594-conjugated anti-rabbit antibodies for 1 h at room temperature and DAPI counterstaining for 5 min at room temperature. For Mbp stainings, blocked sections were incubated with rabbit anti-MBP (1:200, ab40390, Abcam) diluted in 10% goat serum, 1% BSA overnight at 4 °C followed by secondary Alexa fluor^®^ 488-conjugated anti-rabbit antibodies for 1 h at room temperature and DAPI counterstaining for 5 min at room temperature.

Images from the Mbp, Ki67^+^Olig2^+^, CNPase^+^Olig2^+^ and the TUNEL^+^NeuN^+^ stainings were acquired with fluorescent microscopy on a Leica DM6000 B microscope (Leica Microsystems, Wetzlar, Germany) and images from the TUNEL^+^Olig2^+^ stainings were acquired with the Pannoramic MIDI II slide scanner (3DHistech, Budapest, Hungary).

For TUNEL^+^NeuN^+^ stainings, pictures from the parietal cortex and the hippocampus of the ipsilateral hemisphere were taken. Specifically, overview pictures from the posterior parietal cortex and high magnification pictures from the posterior parietal cortex at a fixed distance from the left ventricle and the caudoputamen, as well as overview pictures from the hippocampus and high magnification pictures from the cornu ammonis 1 (CA1) region of the hippocampus at a fixed distance from the hilus of the dentate gyrus and the external capsule were recorded (see Section 3.4). For each animal, a minimum of three high magnification pictures of the parietal cortex and the hippocampus CA1 region were analyzed and double-positive cells were counted by hand. For TUNEL^+^Olig2^+^ stainings, overview pictures from the area where the corpus callosum crosses the hemispheres were taken. For each animal, a minimum of three pictures of the corpus callosum were analyzed and double-positive cells were quantified and calibrated for measured area using ImageJ (NIH). Criteria for cell inclusion was a clear presence of a DAPI^+^ nucleus, together with clear TUNEL^+^NeuN^+^ or TUNEL^+^Olig2^+^ staining, respectively. For each animal, numbers of double-positive cells from all acquired pictures were averaged.

For Ki67^+^Olig2^+^ and CNPase^+^Olig2^+^ stainings, high magnification pictures of the corpus callosum and the external capsule from the ipsilateral hemisphere respectively were taken. For each animal, a minimum of three pictures were analyzed and double-positive cells were counted by hand. CNPase^+^Olig2^+^ cells were counted per field of vision whereas Ki67^+^Olig2^+^ cells were calibrated for measured area using ImageJ. Criteria for inclusion was a clear presence of a DAPI+ nucleus, together with clear Ki67^+^Olig2^+^ and CNPase^+^Olig2^+^ stainings respectively. For each animal, numbers of double-positive cells from all acquired pictures were averaged.

### 2.8. Assessment of Learning and Memory Function by Morris Water Maze

For functional recovery assessment, the learning and memory performance was evaluated 4 weeks post injury using a Morris water maze protocol (Figure 1A) as previously described [38]. For this, Healthy (*n* = 8), PBI (*n* = 13) and PBI+Exo (*n* = 14) animals were transferred to the water maze experimental room and given 3 days to acclimatize in a light cycle of 12 h light and 12 h darkness. The water maze arena consisted of a swimming pool and extra-maze distal cues on the walls surrounding the pool. The pool was filled with dark-colored water and virtually divided into four quadrants. A black platform measuring 13 × 16 cm was placed in the center of the first quadrant 0.5 cm below the water surface, rendering it invisible to the swimming animal. Specific entry zones were defined within the three other quadrants not containing the platform. Over five days, animals performed five training trials per day with the platform in its fixed position to measure learning capacity. During training trials, animals were randomly transferred into one of the three entry zones and were given 90 s to swim to the platform within the first quadrant. If an animal found the platform within 90 s, it was allowed to stay on it for 15 s before the next trial. If the animal did not find the platform within 90 s, it was guided to the platform and was also allowed to stay on it for 15 s. During training trials, the distance the animal swam to reach the platform was recorded. On day 1 and day 5 of the experiment, a probe trial without the platform was performed before and on day 5 also after the training trials. During probe trials, the animals were allowed to swim for 90 s within the pool while their time spent in the quadrant that used to contain the platform was recorded. The first training trial of day 1 was termed “day 0” for the analysis and graphical representation of the training trials. All swimming patterns were recorded and evaluated with the video tracking system EthoVision XT-11 (Noldus Information Technology, Wageningen, The Netherlands).

### 2.9. Statistical Analysis

We used a Mantel–Cox log rank test to investigate the impact of exosome treatment on the survival of animals after PBI. One-way analysis of variance (ANOVA) was used to compare cell death and the number of mature neurons and oligodendrocytes between Healthy, PBI and PBI+Exo animals. Reported *p*-values were Bonferroni-adjusted for multiple comparisons, meaning that the family-wise significance was set to 0.05. For repeated measures over time, we used a two-way ANOVA to analyze the impact of exosome treatment over time for weight gain and learning and memory capacity in Morris water maze testing. To further compare data on a given day between two groups, a paired Student’s t-test was used. The Shapiro–Wilk test was used to test for normality of distributions in all continuous variables. Since all continuous variables were normally distributed, we applied the parametric methods as described. Continuous variables were expressed as mean with 95% confidence interval (CI) unless otherwise stated. Differences between groups were considered significant if (multiplicity adjusted) *p*-values were less than 0.05. Statistical analysis was done using Prism (version 7.0, GraphPad Software, La Jolla, CA, USA).

## 3. Results

### 3.1. MSC-Exosomes Improve Survival after PBI

Weight gain was not significantly altered among all animal groups (Figure 2A). Cumulative survival was significantly reduced in animals with PBI (*n* = 56; Figure 2B) compared to healthy controls (*n* = 8; *p* < 0.0001). Exosome treatment significantly improved cumulative survival after PBI (*n* = 34; *p* = 0.0003). The survival rate at the end of the experiment (P34) was 23.2%, whereas almost half of the PBI animals treated with exosomes survived (41.2%). All healthy control animal survived until P34.

### 3.2. MSC-Exosomes Rach the Brain after Intranasal Administration

Thinking towards future clinical application, we evaluated the intranasal administration of MSC-exosomes as a potential route to deliver them to the brain. We found that 30 min after intranasal administration, IRDye^®^ 800CW-labeled exosomes in PBS appear in the frontal part of the brain including the olfactory bulb and the frontal lobe (Figure 3A). After 3 h, IRDye^®^ 800CW-labeled exosomes were evenly distributed throughout the whole brain. Exosomes also penetrated the brain tissue as IRDye^®^ 800CW-labeled exosomes were found within the deep layers of the brain as seen in the medial view of the left hemisphere (Figure 3B). IRDye^®^ 800CW-labeled exosomes did not appear within the spleen, but a small portion of exosomes was found in the trachea and the gastrointestinal (GI) tract 30 min and 3 h after administration (Appendix A). The specificity of the IRDye^®^ 800CW exosome signal was confirmed with the absence of the IRDye^®^ 800CW signal in Injury animals that received an intranasal administration of PBS only (Figure 3 and Appendix A). The uptake of IRDye^®^ 800CW-labeled exosomes was independent of the injury (Figure 3).

### 3.3. MSC-Exosomes Reduce White Matter Injury after PBI

We used fluorescent TUNEL staining to detect DNA strand breaks of dying cells in combination with nuclear counterstaining and oligodendrocyte marker (Olig2) immunostaining to analyze oligodendrocyte cell death after PBI. Animal models of PBI showed that oligodendrocyte-specific cell death may occur via apoptosis and necrosis [39]. Hence we used TUNEL staining as a sensitive indicator for general cell death with the knowledge that cells dying through either apoptosis or necrosis will display TUNEL signals. We sporadically observed TUNEL^+^Olig2^+^ cells in the corpus callosum in both healthy and PBI animals 24 h after injury (Figure 4A,B), but no differences were observed between the groups (*p* = 0.0783). Exosome treatment did not affect the number of TUNEL^+^Olig2^+^ cells in the corpus callosum of PBI animals (*p* = 0.3876).

We measured Mbp expression to assess the effect of exosome treatment on myelination. Animals with PBI exhibited reduced Mbp expression in the ipsilateral hemisphere at P11 compared to healthy controls (Figure 5A,B). Exosome treatment partially restored Mbp expression after PBI. Specifically, Mbp gene expression (Figure 5A) was reduced by 41% in animals with PBI compared to healthy controls (*p* < 0.0001) and exosome treatment significantly improved Mbp expression by 17% (*p* = 0.0101). Similarly, MBP protein expression (Figure 5B) was reduced by 81% in animals with PBI compared to healthy controls (*p* < 0.0001) and exosome treatment significantly improved MBP ex-pression by 38% (*p* = 0.0156). These findings were confirmed with immunohistochemistry staining showing decreased myelination in the external capsule and corpus callosum of animals with PBI compared to healthy controls, and indicating that exosome treatment restores myelination within the external capsule and corpus callosum after PBI (Figure 5D).

We evaluated the effect of exosome treatment on the number of mature oligodendrocytes by performing double-stainings for developmental stage-specific markers CNPase (more mature oligodendrocytes) and Ki67 (proliferating, more immature oligodendrocytes) together with the general nuclear oligodendrocyte marker Olig2 (Figure 5E–J). The numbers of mature oligodendrocytes was reduced in animals with PBI as they exhibited significantly decreased numbers of CNPase^+^Olig2^+^ cells within the external capsule than healthy control animals (*p* < 0.0001, Figure 5E–G). Exosome treatment significantly rescued the amount of mature oligodendrocyte as illustrated by the increased number of mature CNPase^+^Olig2^+^ cells (*p* = 0.0386). In contrast, animals with PBI did not exhibit more immature oligodendrocytes in the corpus callosum as the number of Ki67^+^Olig2^+^ cells was not increased compared to healthy controls (*p* = 0.0943) and exosome treatment did not affect the number of Ki67^+^Olig2^+^ cells (Figure 5H–J).

### 3.4. MSC-Exosomes Reduce Gray Matter Injury after PBI

Similarly, as for detecting oligodendrocyte-specific cell death, we used TUNEL staining in combination with nuclear counterstaining and neuronal marker (NeuN) immunostaining to detect neuron-specific cell death, as it has also been shown to occur via both apoptosis and necrosis in PBI [17,40]. We found increased numbers of TUNEL^+^NeuN^+^ cells within the subplate zone of the posterior parietal cortex in animals 24 h after PBI (Figure 6A–D) compared to healthy controls (*p* = 0.0001). Exosome treatment significantly decreased the number of TUNEL^+^NeuN^+^ cells within the subplate zone in animals with PBI (*p* = 0.0005). We also found increased numbers of TUNEL^+^NeuN^+^ cells within the CA1 region of the hippocampal formation in animals 24 h after PBI (Figure 6E–H) compared to healthy controls (*p* = 0.0002). Similar as for the cortex, exosome treatment significantly decreased the number of TUNEL^+^NeuN^+^ cells within the CA1 region in animals with PBI (*p* = 0.0034).

We measured Map2 expression to further assess the effect of exosome treatment on neuronal cell counts. Animals with PBI exhibited reduced Map2 expression in the ipsilateral hemisphere at P11 compared to healthy controls and exosome treatment partially restored Map2 expression after PBI (Figure 6I,J). More specifically, Map2 gene expression (Figure 6I) was reduced by 25% in animals with PBI compared to healthy controls (*p* = 0.0021) and exosome treatment significantly improved Map2 expression after PBI by 19% (*p* = 0.0043). Similarly, MAP-2 protein expression (Figure 6J) was reduced by 85% in animals with PBI compared to healthy controls (*p* < 0.0001) and exosome treatment significantly improved MAP-2 expression after PBI by 33% (*p* = 0.0002).

### 3.5. MSC-Exosomes Improve Functional Recovery after PBI

Long-term neurofunctional outcome after PBI was evaluated using a Morris water maze test (Figure 7A) four weeks after injury. Learning ability was observed in all animal groups, as indicated by a steep decrease of total distance swum until reaching the platform between the first training trial (designated day 0) and the next four training trials of day 1 (designated day 1, Figure 7B). Healthy control animals (*n* = 8) showed significant better learning performance than animals with PBI (*n* = 13) on training days 2 (*p* = 0.0208), 3 (*p* = 0.0013) and 5 (*p* < 0.0001). PBI animals receiving exosome treatment (*n* = 14) performed significantly better in the learning assessment than untreated PBI animals on training days 3 (*p* = 0.0246) and 5 (*p* = 0.0106).

Short-term memory was similar among all animal groups, as indicated by an increase in the time spent in the platform quadrant between the first and second probe trial on day 5 (Figure 7C). Long-term memory was significantly reduced in PBI animals compared to healthy controls (*p* = 0.0489), as indicated by the decrease in the time spent in platform quadrant during the first probe trial on day 5 (5.1). Exosome treatment did not rescue long-term memory impairment after PBI (*p* = 0.6106).

## 4. Discussion

Our study provides evidence that MSC exosomes have neuroprotective effects as they reduce both white and gray matter alterations in PBI. We demonstrated that intranasally administered exosomes reach the brain where they not only reduce neuronal cell death but also increase mature oligodendrocyte counts, thereby improving learning ability after PBI.

### 4.1. White Matter Alterations in PBI

Evidence from our study indicates that white matter injury in PBI does not result from oligodendrocyte-specific cell death and is more likely to be caused by impaired oligodendrocyte maturation. These findings are in line with previous studies documenting no oligodendrocyte-specific cell death in white matter injury. For instance, our previous study showed that arrested oligodendrocyte maturation is most likely responsible for the myelination deficits in neonatal rats with preterm brain injury [23]. Similarly, in an animal model of diffuse white matter injury, combined fetal inflammation and postnatal hypoxia caused myelination and neurobehavioral deficits resulting from arrested oligodendrocyte maturation rather than from oligodendrocyte-specific cell death [13]. In this model, animals with brain injury exhibited impaired oligodendrocyte maturation as illustrated by decreased numbers of mature CNPase^+^Olig2^+^ and increased numbers of Ki67^+^Olig2^+^ oligodendrocytes within the cortex and corpus callosum respectively. While our study also found decreased numbers of mature CNPase^+^Olig2^+^ oligodendrocytes, our model did not replicate the increased amounts of immature, proliferating Ki67^+^Olig2^+^ oligodendrocytes. Interestingly, also neuropathological studies confirmed arrested oligodendrocyte differentiation as the underlying cause of altered myelination in premature infants as they found no oligodendrocyte-specific cell death but increased amounts of maturation-arrested oligodendrocytes [12]. In contrast, other studies described substantial oligodendrocyte-specific cell death as contributing mechanisms for the observed white matter alterations in postmortem brains of premature newborns [10,11].

### 4.2. Gray Matter Alterations in PBI

Our study showed that gray matter injury in PBI likely results from neuron-specific cell death. These findings are in line with studies showing substantial gray matter cell death after PBI. For example, in an animal model of neonatal hypoxia-ischemia, widespread subcortical and periventricular cell death was observed and—similar to our findings—dying cells were identified as subplate neurons [17,18]. Subplate neurons are transient neurons occurring in the developing preterm cerebral white matter and similar to immature oligodendrocytes, subplate neurons are especially vulnerable to hypoxia-ischemia [17] due to their high expression of NMDA and AMPA receptors [42], which are linked to excitotoxicity in the pathogenesis of PBI. In a similar animal model of PBI, neuron-specific cell death has been shown within the deep layer of the cortex close to the subplate zone. [43]. Moreover, analogous to our findings of hippocampal neuronal cell death, neuron-specific cell death within the CA1 and CA3 region of the hippocampus was observed 24 h after neonatal hypoxia-ischemia injury [16]. Hippocampal neurons within the CA1 region are especially known to be vulnerable to hypoxic-ischemic injury. Similar gray matter injuries have also been found in postmortem brains of prematurely born infants, where significant losses of axons and gamma-aminobutyric acid (GABA)-expressing subplate neurons were observed [11]. A contrasting study found no acute or delayed neuronal death after transient cerebral hypoxia-ischemia in preterm sheep [19].

### 4.3. Long-Term Neurodevelopmental Deficits after PBI

We observed significant differences in learning ability and memory formation in animals four weeks after PBI, which are likely to be related to the observed injuries within the hippocampus acutely after birth. The hippocampus is known for its crucial role in memory formation [44] and structural deficits to the hippocampus following preterm birth have been shown to affect episodic memory abilities during childhood and into early adulthood [2]. While the hippocampus is crucial for memory formation, hippocampo-cortical and cortico-cortical back projections are required for the subsequent retrieval of these memories [44]. Further, the performance in the Morris water maze test is dependent on the visuospatial working memory [45] which is known to be impaired in patients with lesions within the posterior parietal cortex [46]. The requirement of functional cortical back projections and intact posterior parietal cortex for memory formation provides an interesting lead regarding the mechanisms through which neuronal cell death within the subplate zone of the posterior parietal cortex could also contribute to memory and learning deficits in PBI. However, more detailed analyses are needed to directly relate these findings to the cell death of subplate neurons within the posterior parietal cortex and to definitely assess the contribution of cortical gray matter injury to neurodevelopmental deficits in PBI. Even though we found a significant difference in learning ability and memory formation between healthy controls and animals with PBI, PBI animals still demonstrated some learning capacity as observed by their increasing performance in the water maze assay over time. This is in line with the present clinical situation, in which PBI in the context of preterm birth is generally associated with subtle cognitive and memory impairments, rather than severe mental retardation [47].

### 4.4. MSC-Exosomes as a Treatment for PBI

Our study revealed that oligodendroglial dysmaturation and neuron-specific cell death are important elements within the pathogenesis of preterm PBI which are contributing to the observed long-term neurodevelopmental deficits. This implicates a specific need for neuroprotective therapies which should prevent neuronal cell death while simultaneously supporting oligodendroglial maturation, thereby preventing long-term neurodevelopmental deficits. We hereby provide evidence that MSC-exosomes have the potential to fulfill these requirements.

We found that MSC exosomes are able to prevent neuron-specific cell death within the posterior parietal cortex as well as within the hippocampus. This is in line with previous findings of our group, where MSC exosomes prevented neuron-specific apoptosis during an in vitro-model of regeneration [32]. Similarly, another study in an inflammatory model of PBI showed an increased amount of TUNEL positive cells within the cortex which was reduced upon MSC-derived extracellular vesicle treatment [26]. The same study further described deficits in adaptive memory function after PBI which were improved upon treatment with MSC-derived extracellular vesicles. Other studies also described similar anti-apoptotic, neuroprotective and neuroregenerative effects of MSC-derived extracellular vesicles in comparable adult models of acute cerebral tissue damage such as stroke and traumatic brain injury [27,28,48]. One of these studies also evaluated the learning ability and memory performance of rats after traumatic brain injury using the Morris water maze assay and found improvements in memory capacity upon the treatment with MSC-exosomes [27]. In our model, MSC-exosome treatment resulted in an improvement of spatial learning, but not of the long-time memory. We speculate that the time spent in the correct quadrant as a measure of spatial long-time memory is not only dependent on hippocampus-associated memory formation but also on the types of swimming paths performed [49]. For future studies, a more in-depth analysis of allo- versus egocentric swimming paths might help to distinguish adopting strategies independent of distal visual cues from memory-dependent orientation.

We also found evidence that MSC-exosomes support oligodendroglial maturation. This is in line with other studies describing that MSC-derived extracellular vesicles attenuate hypomyelination using Mbp as a marker for myelination processes [26]. While Mbp is not expressed by immature oligodendrocytes, it is expressed and secreted by mature oligodendrocytes [50]. Hence, it is likely that the MSC-derived exosome treatment supported the oligodendrocyte maturation after PBI in this study. With regards to a potential future clinical application, we found that MSC-exosomes can be administered in a minimally invasive way through the nose to effectively deliver them to the brain. This is in agreement with studies describing effective MSC-exosome delivery through the nose to the brain to treat various CNS diseases such as Parkinson’s disease [51], epilepsy [52], autism spectrum disorders [53] and neuroinflammatory diseases in general [54]. Whereas in most of these studies and also in ours, the observed kinetics of exosome translocation to the brain are most consistent with the previously described trigeminal and the olfactory routes [55], further studies are needed to clarify the exact route by which exosomes travel from the nose to the brain [56,57].

To support the stringency of our findings, future studies would be beneficial to further characterize the dying neurons within the subplate zone of the posterior parietal cortex to confirm their subplate neuronal identity and to differentiate them from forkhead box protein 1 (foxp1)-expressing neighboring neurons from cortical layers II–V [18]. Additionally, a very recent study showed that the loss of GABA-ergic interneurons within the prefrontal cortex in a novel mouse model of preterm brain injury lead to specific working memory deficits and neurobehavioral deficits which parallel human psychopathologies seen in preterm birth survivors [58]. Further investigations would be required to analyze whether there are also GABA-ergic interneurons among the dying cells within the parietal cortex.

Our previous study suggests that the immunomodulatory effects on microglia [35] indirectly and the anti-apoptotic effects on neuronal cells [32] directly contribute to the neuroprotective effects of MSC-exosomes observed in this study. To complete our understanding of the observed neuroprotective effects of MSC-exosomes, key exosomal molecules and involved mechanisms should be identified. There is growing evidence for the importance of specific microRNA (miRNA) in brain development and PBI [59]. MicroRNAs have key functions in oligodendroglial maturation, myelination and apoptosis. We have previously shown by a real-time PCR array that MSC-exosomes contain many miRNAs targeting pro-apoptotic genes, including let-7 family [32]. To get a detailed profile of the exosomal miRNA cargo, Illumina NextSeq Sequencing has been done. Preliminary results indicate that several anti-apoptotic miRNAs are highly expressed in MSC-exosomes, including let-7, miR-21, and miR-22, verifying our previous findings [32,60,61]. Furthermore, the preliminary sequencing results indicate that levels of exosomal miR-199a-5p and miR-145, having critical roles in the maturation of oligodendrocytes and the generation of myelin, are high. We, therefore, speculate that the above-mentioned miRNA have key functions in the neuroprotective impact of MSC-exosomes. However, the detailed characterization of the exosomal miRNA cargo and functional validation will be the subject of a future study.

In the present study, we show that MSC-exosomes have a protective effect on white and gray matter injury in a model of PBI. In accordance with our previous experience with the model [35] and since we also performed TUNEL assays where the animals were sacrificed already 24 h after LPS injection, an experimental setup with small time intervals between exosome application and injury induction was used. Therefore, the exosomes were transplanted during injury induction, meaning that the exosomes were applied after LPS injection, but immediately before carotid ligation, followed by hypoxia. We are aware that relating to the clinical situation it is essential to evaluate post-injury time points for treatment as a preventive application is only of limited clinical utility. Studies testing different and/or repeated doses of exosomes to determine their ideal therapeutic window as well as examining different time points for the treatment remain to be done. Further experiments are needed to characterize the pharmacological properties of MSC-exosomes as a treatment for preterm injury. However, using intranasal administration, we have already established an application route with an optimal balance between effective delivery and invasiveness.

## 5. Conclusions

In conclusion, we showed that MSC exosomes applied prior ischemia very significantly prevented perinatal brain injury and propose that MSC exosomes represent a promising strategy to prevent preterm PBI in human newborns. Their capacity to prevent gray and white matter alterations and especially their ability to improve long-term neurodevelopmental outcome as well as their feasibility to be administered in a minimally invasive yet effective, intranasal manner renders MSC exosomes a novel prevention strategy.

## Figures and Tables

**Figure 1 cells-08-00855-f001:**
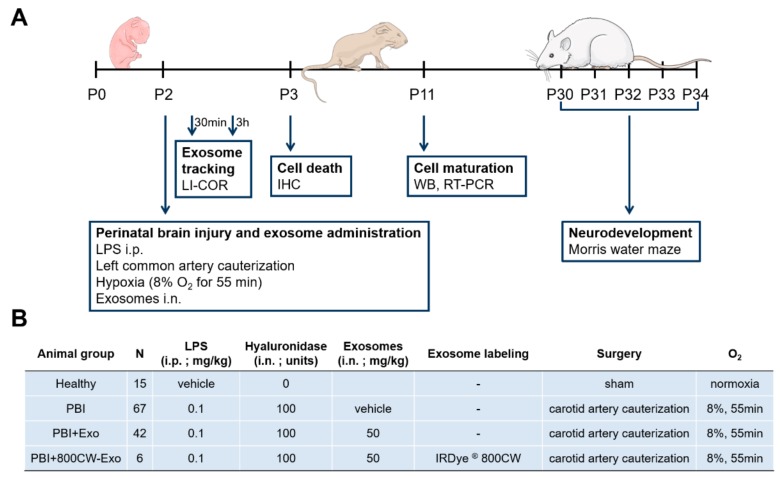
Rat model of perinatal brain injury. (**A**) Schematic representation of the experimental outline. Perinatal brain injury was induced in postnatal day 2 (P2) rat pups by i.p. injection of LPS, followed by left common carotid artery cauterization 2 h later and exposure to hypoxia for 55 min. Native or labeled exosomes were administered intranasally (i.n.) immediately before the cauterization of the carotid artery. Labeled 800CW-exosomes were tracked 30 min and 3 h after administration throughout the animal’s body using the LI-COR Odyssey imaging system. Oligodendrocyte- and neuron-specific cell death was measured on P3 using immunohistochemistry (IHC). Mature neuronal and oligodendroglial cell count was assessed on P11 using Western blot (WB) and reverse transcription PCR (RT-PCR). Long-term neurodevelopment was evaluated between P30 and P34 using Morris water maze assay. (**B**) Detailed overview of the four experimental animal groups.

**Figure 2 cells-08-00855-f002:**
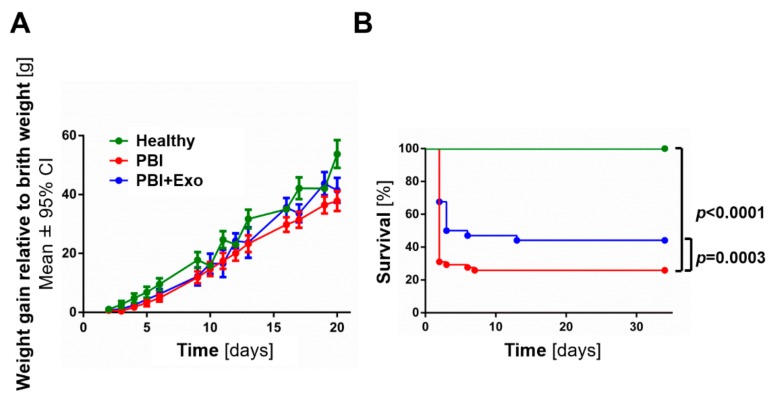
Weight gain and survival after perinatal brain injury (PBI). Weight gain (**A**) and survival (**B**) of preterm rats with PBI with or without exosome (Exo) treatment. Error bars illustrate mean ± 95% confidence interval (CI) of at least eight different animals.

**Figure 3 cells-08-00855-f003:**
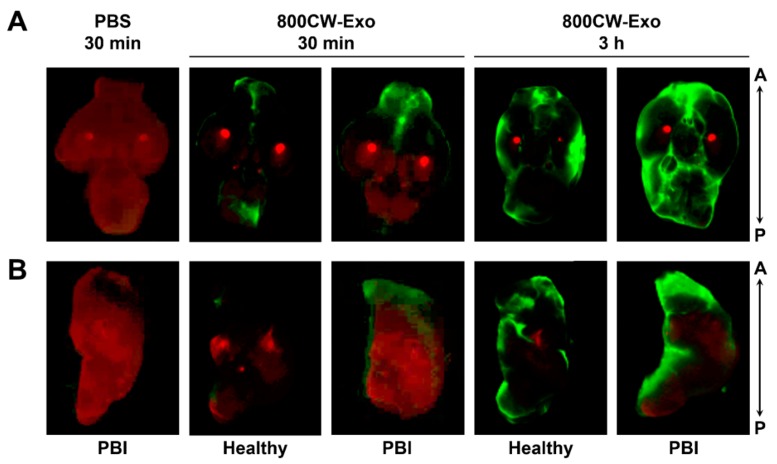
Intranasal administration of mesenchymal stromal cell-derived exosomes. Representative images of brains from healthy rats or rats with PBI 30 min and 3 h after intranasal administration of either PBS or IRDye^®^ 800CW-labeled exosomes (green) in PBS. In (**A**), the whole brain was imaged from an inferior view and in (**B**), the brain was sagitally cut and the left hemisphere was imaged from a medial view to see inside the brain parenchyma. Background autofluorescence from brain tissue is depicted in red. A: anterior, P: posterior.

**Figure 4 cells-08-00855-f004:**
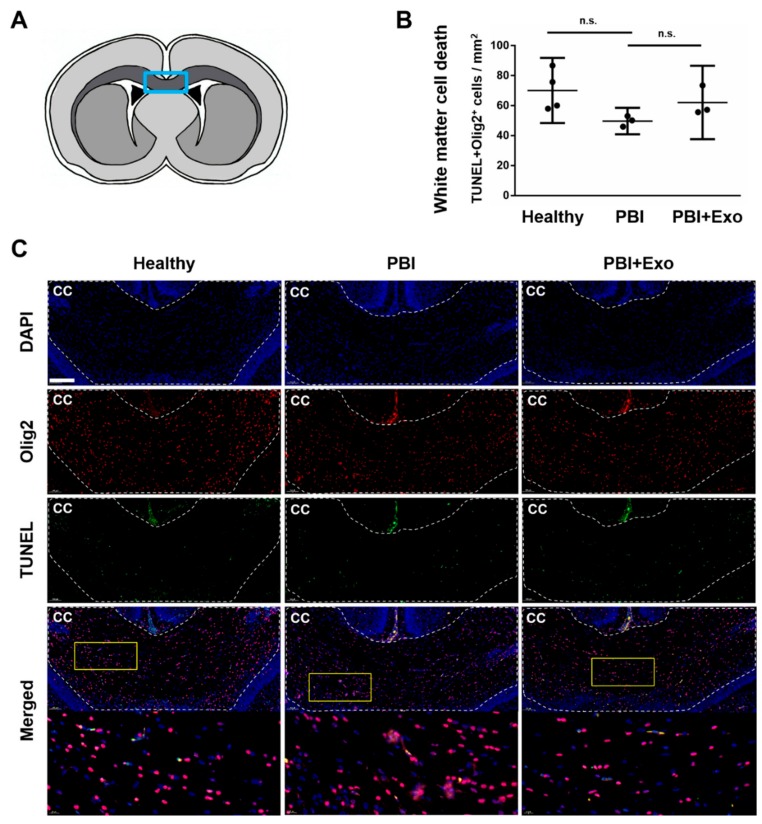
Oligodendrocyte cell death after perinatal brain injury (PBI). (**A**) Scheme of coronal brain section with the position of the area used for quantification (light blue box). Quantification (**B**) and representative overview images (**C**) of the corpus callosum (cc) double stained for DNA fragmentation marker terminal deoxynucleotidyl transferase dUTP nick end labeling (TUNEL) and Olig2 showing Olig2^+^TUNEL^+^ cells in the cc of healthy rats and of rats 24 h post brain injury with and without exosome (Exo) treatment. Yellow boxes indicate positions of higher magnification images shown below. Scale bar: 200µm. Error bars illustrate mean ± 95% confidence interval (CI) of at least three different animals. n.s. *p* > 0.05.

**Figure 5 cells-08-00855-f005:**
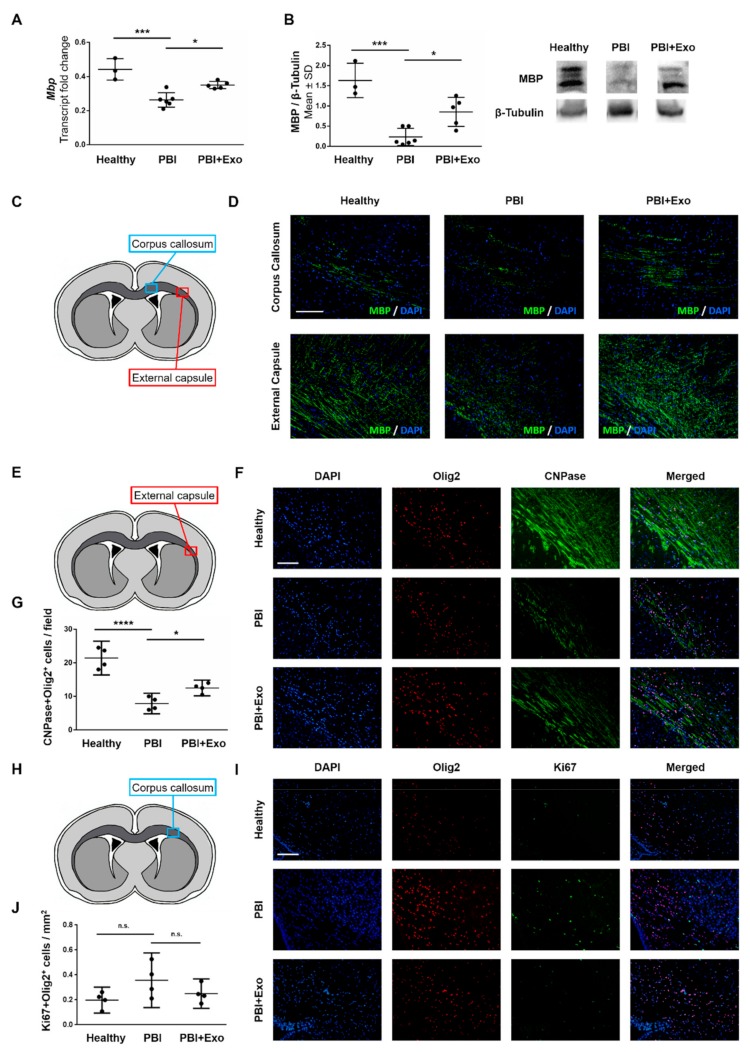
MSC-exosomes reduce white matter alterations after perinatal brain injury (PBI). Oligodendroglial markers were measured in healthy rats and in rats with PBI with or without exosome (Exo) treatment on postnatal day 11 (P11). Quantification of Mbp gene (**A**) and protein (**B**) expression in the brain parenchyma. (**C**) Scheme of a coronal brain section indicating the positions of the areas in the corpus callosum (light blue box) and the external capsule (red box) taken for analysis of Mbp expression and representative images (**D**) of Mbp expression in the corpus callosum and the external capsule. Schematic representation of the area analyzed (**E**, red box), representative images (**F**) and quantification (**G**) of CNPase^+^Olig2^+^ cells in the external capsule. Schematic representation of the area analyzed (**H**, light blue box), representative images (**I**) and quantification (**J**) of Ki67^+^Olig2^+^ cells in the corpus callosum. Scale bars: 200µm. Error bars illustrate mean ± 95% confidence interval (CI) of at least three different animals. * *p* ≤ 0.05, *** *p* ≤ 0.001, **** *p* ≤ 0.0001, n.s. *p* > 0.05.

**Figure 6 cells-08-00855-f006:**
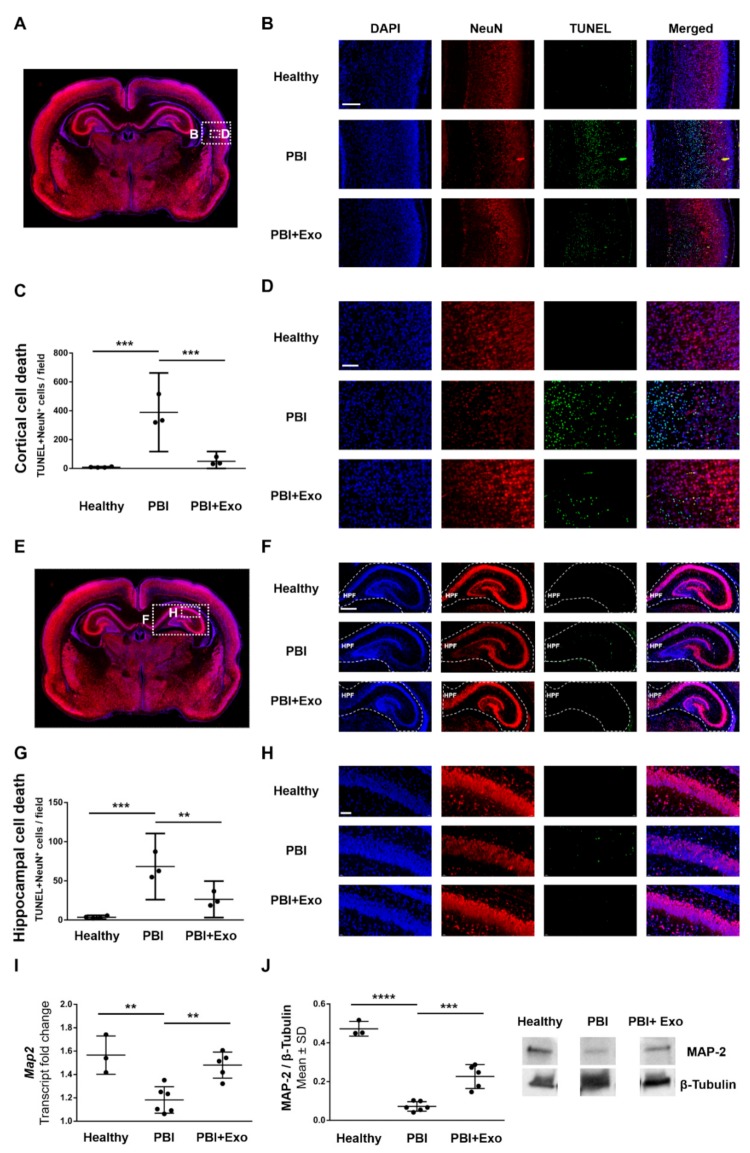
MSC-exosomes prevent gray matter alterations after perinatal brain injury (PBI). Neuronal cell death was assessed in healthy rats and in rats 24 h post brain injury with or without exosome (Exo) treatment (**A**–**H**). (**A**) Overview of rat brain coronal sections with inserts indicating the region in the posterior parietal cortex for images in (**B**,**D**). (**B**) Representative overview images of the posterior parietal cortex region double-stained for DNA fragmentation marker TUNEL and NeuN. Scale bar: 200 µm. Quantification (**C**) and representative high magnification images (**D**) of NeuN^+^TUNEL^+^ cells in the posterior parietal cortex. Scale bar: 100 µm. (**E**) Overview of rat brain cor-onal sections with inserts indicating the regions of the hippocampus for images in (**F**,**H**). (**F**) Representative overview images of the hippocampal formation (HPF) region double-stained for DNA fragmentation marker TUNEL and NeuN. Scale bar: 400 µm. Quantification (**G**) and representative high magnification images (**H**) of NeuN^+^TUNEL^+^ cells in the cornu ammonis (CA) 1 region of the HPF. Scale bar: 100 µm. Quantification of microtubule-associated protein 2 (Map2) gene (**I**) and protein (**J**) expression in the brain parenchyma on postnatal day (P) 11. Error bars illustrate mean ± 95% confidence interval (CI) of at least three different animals. ** *p* ≤ 0.01, *** *p* ≤ 0.001, **** *p* ≤ 0.0001.

**Figure 7 cells-08-00855-f007:**
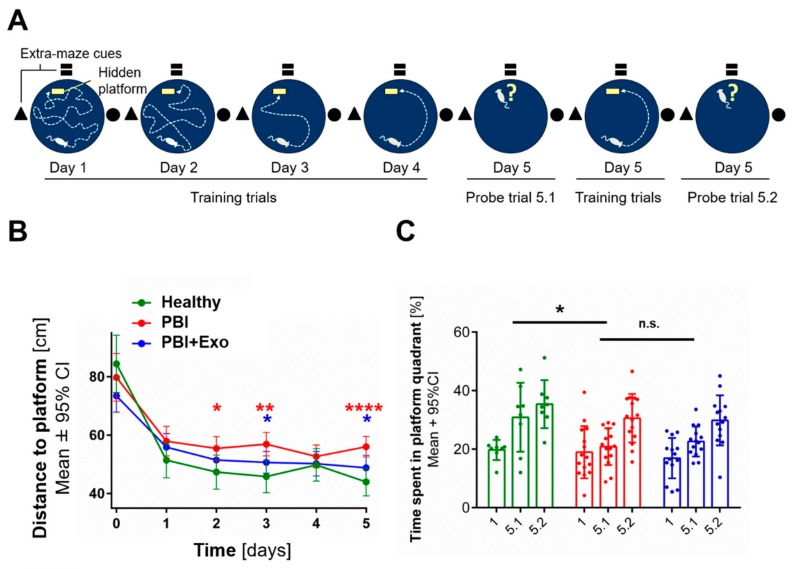
MSC-exosome rescue neurodevelopmental outcome after PBI. (**A**) Experimental overview of the Morris water maze protocol used to assess learning and memory performance between P30 and P34 (scheme inspired by Kipnis et al.) [41] (**B**) Quantification of learning behavior assessed by swimming distance until finding the platform within training trials on five consecutive days. Red stars indicate significant differences between Healthy and PBI animals, blue stars show significance of the exosomes’ therapeutic effects (PBI+Exo vs. PBI). (**C**) Quantification of memory formation assessed with probe trials evaluating time spent in target quadrant after removing the platform on days 1 and 5. Error bars illustrate mean ± 95% confidence interval (CI) of at least eight different animals. * *p* ≤ 0.05, ** *p* ≤ 0.01, **** *p* ≤ 0.0001, n.s. *p* > 0.05.

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
