# Peer review of "Intranasally Administered Exosomes from Umbilical Cord Stem Cells Have Preventive Neuroprotective Effects and Contribute to Functional Recovery after Perinatal Brain Injury"

_cells, 2019, doi:10.3390/cells8080855_

Round 1

Reviewer 1 Report

The authors investigated the neuroprotective role of mesenchymal stromal cell (MSC)-derived exosomes in preventing neurodegeneration occurring after a perinatal brain injury (PBI) in in vivo models. They isolated exosomes from human Wharton's jelly MSC supernatant and administered them intranasally at P2 in a rat model of PBI. Using different techniques (from molecular biology to immunohistochemistry to imaging to behavioural tests) they demonstrated that exosomes were able to migrate throughout the whole brain after 3 hours of administration. The authors observed a recovery of neuron cell counts, a normalization of myelination and oligodendrocyte maturation, which led to an improvement in learning ability of treated animals.

The present findings and the indication of MSC-derived exosome administration as novel treatment strategy in PBI confirm in vivo the authors' previous data on neuroprotective exosome effects observed in vitro on cell lines.

This is an interesting and well-written study. The methodology is clearly detailed and covers different aspects of the research. The conclusions are supported by reliable experimental data.

Some comments to improve the manuscript:

- Based on literature data and authors' previous in vitro results, a targeted RNA and/or protein profile should be done in order to identify the molecules contained in the exosomes that might exert the neuroprotective effects. These results will reinforce the validity and the importance of the present study.

- Exosomes were administered between LPS injection and the ligation at P2. It is not clear the timing of administration; one it would expect the administration after the PBI induction as therapeutic therapy. Even if the authors are conscious that a preventive therapy has a limited clinical utility, they should explain in more details why they decided the time point in the present settings of experiments.

- How many investigators assessed the sections? Were the blinded to the diagnosis? Were the pictures selected randomly? Could the authors include in figures 4 and 5 an insert indicating the area where the counts were done.

Author Response

We would like to thank the reviewers for taking their time to provide constructive comments and suggestions.

Please see below for our detailed responses.

Point 1: Based on literature data and authors' previous in vitro results, a targeted RNA and/or protein profile should be done in order to identify the molecules contained in the exosomes that might exert the neuroprotective effects. These results will reinforce the validity and the importance of the present study.

Response 1: We thank the reviewer for this valuable comment. We completely agree that key molecules involved in the neuroprotective impact of MSC-exosomes should be identified, as we also stated in the Discussion of the manuscript (p.17, l. 570). Mature microRNAs (miRNA) are promising candidates, as they are highly conserved across species, and have key functions in oligodendroglial maturation, myelination, and apoptosis. The growing evidence of the importance of specific miRNA in brain development and perinatal brain injury is extensively described by the recent review of Cho KHT et al., Front Physiol, 2019.

In our previous publication from 2018 (Joerger-Messerli et al., Cell Transplant, 2018) we already showed by real-time PCR that MSC-exosomes contain miRNA targeting pro-apoptotic genes, including mature Let-7. However, to get an all-embracing exosomal miRNA profile, we performed genome-wide profiling of MSC-derived exosomes. Small RNA libraries have been prepared, followed by Illumina NextSeq Sequencing to identify miRNAs. The detailed characterization of the exosomal miRNA cargo will be the subject of a separate manuscript, as the inclusion of these data would go beyond the scope of this manuscript. Preliminary results indicate that, surprisingly, the levels of miR-219, miR-338, and miR-138, which are important regulators of oligodendrocyte differentiation, were very low to not measurable. In contrast, the levels of exosomal miR-199a-5p and miR-145, two miRNA having critical roles in the maturation of oligodendrocytes and the generation of myelin, were highly expressed. Furthermore, exosomal miR-146a, promoting oligodendrogenesis during perinatal brain injury by resolving arrested oligodendrocyte maturation, was high as well.

Concerning miRNA with antiapoptotic functions, we found that all members of the Let-7 were highly expressed in MSC-exosomes, validating our findings form 2018. Furthermore, miR-21 (Buller et al., FEBS J, 2010) and miR-22 (Jovicic et al., PlosOne, 2013) exerting their neuroprotective function by inhibiting apoptosis, were highly expressed in exosomes.

To conclude, we speculate that exosomal miR-199a-5p, miR-145 and miR-146a are promising candidates to improve oligodendrocyte maturation and myelination in perinatal brain injury. Exosomal let-7, miR-21, and miR-22 could be (at least partly) responsible for the anti-apoptotic effect of MSC-exosomes. However, a detailed miRNA cargo profile of MSC-exosomes will follow in a separate manuscript. The preliminary results are now included in the Discussion (p. 17, l. 571-582).

Point 2: Exosomes were administered between LPS injection and the ligation at P2. It is not clear the timing of administration; one it would expect the administration after the PBI induction as therapeutic therapy. Even if the authors are conscious that a preventive therapy has a limited clinical utility, they should explain in more details why they decided the time point in the present settings of experiments.

Response 2: We totally agree with the reviewer that the exosome administration post ligation is very interesting concerning the clinical situation.

In the present study, however, we aimed to determine wether MSC-exosomes have a beneficial effect on white and gray matter injury in a rat model of PBI at all. For the TUNEL stainings, the animals have been sacrificed already 24 h post LPS administration – meaning about 22h after exosome administration and 16 h post hypoxia and ischemia. As we wanted to be sure that the exosomes had enough time to exert their effects, we have chosen an experimental setup with a small time interval between the exosomes administration and injury induction. Therefore, we decided to apply the exosomes during the injury induction, meaning after LPS injection, but before carotid ligation and hypoxia.

Having shown now the neuroprotective effect of MSC-exosomes, the determination of a more clinically relevant time point of administration and dosage of exosomes is subject of the current research. This point is now discussed in the manuscript (p. 17, l. 583-590)

Point 3: How many investigators assessed the sections? Were the blinded to the diagnosis? Were the pictures selected randomly? Could the authors include in figures 4 and 5 an insert indicating the area where the counts were done.

Response 3: The sections have been assessed by one investigator. Before analysis, the slides have been anonymized and analyzed in a blinded way. The images were randomly selected by staining and imaging every third brain slice in the region of interest. This information is now included in the Materials and Methods (p. 5, l. 204-206). The areas where the cell counts were done are now indicated in a schematic way in both Figures 4 and 5. The images in the panels 5E and 5G of the previous version (now 5F and 5I, respectively) were mirrored to match the location of the areas as indicated in the schemes. We chose to do so because the original images were taken from slides that were mounted the other way round (ipsilateral hemisphere on the left), while the slides taken for Figures 4, 5C (now 5D) and 6 were mounted with the ipsilateral hemisphere on the right side.

Reviewer 2 Report

The manuscript entitled Intranasally administered exosomes from umbilical cord stem cells have neuroprotective effects and contribute to functional recovery after perinatal brain injury by Thomi et al is an original research manuscript that has interestingly demonstrated that intranasally administered MSC-derived exosomes can enter the brain and if administered after perinatal brain injury (PBI) have protective effects on white matter and neuronal population. There were also behavioral studies that attempted to answer the question whether these MSC-derived exosomes after PBI improved outcomes. The investigators clearly demonstrate that these MSC-derived exosomes can enter the brain when administered intranasally. While this study is very interesting and provides good evidence of the benefits of the MSC-derived exosomes, there are several issues that need to be addressed.

It is not clear why the exosomes were administered prior to the carotid ligation? It would have been interesting to have a group where the exosomes were administered after HI event.

In the methods section, page 6 the authors need to describe approximate location of white matter in relation to Bregma coordinates.  

Why was survival in the PBI group so low? Could this introduce bias in your results?

No mention of sex as a biological variable. If both sexes are analyzed, then this should be stated.

Does PBI alter the distribution and absorption of labeled exosomes? The investigators did a terrific experiment demonstrating that the IRDye-labeled exosomes entered the brain and the distribution at 30 min and 3 hours.  

Images in Figure 4B are small and difficult to visualize. A small insert demonstrating double labeled cells would also be beneficial.

The investigators describe “exosome treatment on oligodendrocyte maturation…” (ex: page 9, line 333) However, what they are really demonstrating is the number of mature oligodendrocytes. To demonstrate that the exosomes accelerated maturation, then labeling with BrdU pulse would be necessary.   The same would apply for neurons.

Clarification of Figure 7B: Is the significant difference in the PBI+exosome group in comparison to PBI or to healthy controls? Based on the figure y-axis, difficult to visualize and may need different scale.

In general, the results in figure 7 demonstrate that the PBI group did learn but were significantly worse than the healthy group. The PBI+exosome group did learn better as is evident on day 3 and 5. However, in the memory test, as indicated by the time spent in platform quadrant, there was no difference in the PBI+exosome compared to the PBI only group. This interpretation needs to be clarified.   

Author Response

We would like to thank the reviewers for taking their time to provide constructive comments and suggestions.

Please see below for our detailed responses.

Point 1: It is not clear why the exosomes were administered prior to the carotid ligation? It would have been interesting to have a group where the exosomes were administered after HI event.

Response 1: We totally agree with the reviewer that the exosome administration post ligation is very interesting concerning the clinical situation.

In the present study, however, we aimed to determine whether MSC-exosomes have a beneficial effect on white and gray matter injury in a rat model of PBI at all. For the TUNEL stainings, the animals have been sacrificed already 24 h post LPS administration – meaning about 22h after exosome administration and 16 h post hypoxia and ischemia. As we wanted to be sure that the exosomes had enough time to exert their effects, we have chosen an experimental setup with a small time interval between the exosomes administration and injury induction. Therefore, we decided to apply the exosomes during the injury induction, meaning after LPS injection, but before carotid ligation and hypoxia.

Having shown now the neuroprotective effect of MSC-exosomes, the determination of a more clinically relevant time point of administration and dosage of exosomes is subject of the current research. This point is now discussed in the manuscript (p. 17, l. 583-590)

Point 2: In the methods section, page 6 the authors need to describe approximate location of white matter in relation to Bregma coordinates.

Response 2: Thank you for this useful comment. The coordinates of the coronal brain sections used in immunohistochemistry are now indicated relative to Bregma for the corpus callosum (cc), external capsule (ec) and hippocampus at P3 and cc + ec for P11 animals (p. 5, l. 201-204).

Point 3: Why was survival in the PBI group so low? Could this introduce bias in your results?

Response 3: We thank the reviewer for this valuable question. Compared to our previous experiences with a neonatal rat PBI model where we had a 24h interval between the LPS administration and the hypoxia/ischemia, we had indeed a higher loss rate. Since the cauterization of the carotid artery was performed by the same investigator who has a long-standing experience with the method and hypoxia was performed following our standard protocol, we attribute the lower survival to the time change in LPS administration. The time change in LPS was inevitable because of two separate reasons: First, we were interested in short-time alterations that are linked to the acute phase of the damage (see also our response to question 1 / question 2 Reviewer 1). Second, LPS administration 24h prior to hypoxia/ischemia has been proven to be protective in PBI (Dhillon et. al., Dev Neurosci, 2015; Lin et al., Pediatr Res, 2009) whereas an LPS administration 1 to 4 hours prior to hypoxia/ischemia seems to potentiate the injury (Lehnardt et al., Proc Natl Acad Sci USA, 2003; Wang et.al., Pediatr Res, 2010, Dhillon et. al., Dev Neurosci, 2015). We are constantly focusing on improving the survival of the pups in our experiments. Survival of the pups also very much depends on maternal care. Following our experience with the study presented here, we have now recently introduced a new handling protocol with the aim to increase the pregnant dams’ habituation to the experimenter and therefore reduce stress in the animals. We cannot completely exclude that the low survival in the PBI group has introduced some bias into our results. Nevertheless, we presume that PBI is the main reason for the low survival of affected animals. This means that we probably miss animals with more severe brain damage in the PBI group. If we were able to include those animals, the differences to the healthy animals’ group would be bigger and more significant. Since we evaluate brain damage in the (less damaged) survivors only, we rather underestimate the damage over all animals. Exosome-treatment of PBI animals improved survival (Figure 2B), indicating that the treatment rescued some of the animals who’s survival was doubtful. Pups intranasally treated with exosomes also swallow and aspirate some of the exosomes. Common (co-) morbidities following preterm birth are necrotizing enterocolitis (NEC) and bronchopulmonary dysplasia (BPD). Therefore, the exosome treatment might also affect NEC and BPD outcomes, further influencing survival.

Point 4: No mention of sex as a biological variable. If both sexes are analyzed, then this should be stated.

Response 4: We agree with the reviewer that we did not explain how we handled sex as a biological variable. In our study, both sexes have been analyzed. This is now stated in the Materials and Methods (p. 3, l. 126). For the Morris water maze assay, both sexes have been analyzed separately as well. However, there have been no differences between males and females. For the other experiments, the animal groups have been too small to analyze both sexes separately.

Point 5: Does PBI alter the distribution and absorption of labeled exosomes? The investigators did a terrific experiment demonstrating that the IRDye-labeled exosomes entered the brain and the distribution at 30 min and 3 hours.

Response 5: We thank the reviewer for the interesting question and his approval on our experiment. The absorption and distribution profiles of IRDye-labeled exosomes were similar in healthy control and injured animals. To illustrate this, we now also included brain images of IRDye-exosome-treated animals taken at the same time points after intranasal administration and made a note in the corresponding results section (p. 7, l. 321-322),

Point 6: Images in Figure 4B are small and difficult to visualize. A small insert demonstrating double labeled cells would also be beneficial..

Response 6: We agree with the reviewer that the images were a bit small. The images in Figure 4B are now slightly (+37%) bigger. Matching higher resolution images were added for merged channels. Since we did not have matching high-resolution images available for the brain slices previously chosen for the PBI and PBI+Exo groups, those images were replaced by new images representing brains from different animals in the same series of experiments.

Point 7: The investigators describe “exosome treatment on oligodendrocyte maturation…” (ex: page 9, line 333) However, what they are really demonstrating is the number of mature oligodendrocytes. To demonstrate that the exosomes accelerated maturation, then labeling with BrdU pulse would be necessary.   The same would apply for neurons.

Response 7: The reviewer is absolutely right. We changed the wording in the manuscript accordingly (p. 1, l. 29-30, 36-37; p. 4, l. 146-147; p. 5, l. 217-218; p. 6, l. 285-285; p. 9, l. 362-372; p.14, l. 460-462).

Point 8: Clarification of Figure 7B: Is the significant difference in the PBI+exosome group in comparison to PBI or to healthy controls? Based on the figure y-axis, difficult to visualize and may need different scale.

Response 8: Thank you, the color code indicating significant differences is now clarified in the figure legend. We agree that overlapping confidence intervals are somewhat difficult to see, but this is hardly influenced by the scale of the y-axis. The clarification of the significance levels should now lead to a better understanding of the figure.

Point 9: In general, the results in figure 7 demonstrate that the PBI group did learn but were significantly worse than the healthy group. The PBI+exosome group did learn better as is evident on day 3 and 5. However, in the memory test, as indicated by the time spent in platform quadrant, there was no difference in the PBI+exosome compared to the PBI only group. This interpretation needs to be clarified.

Response 9:

We agree that, in our experimental setup, the time spent in the quadrant is a less sensitive measure compared to the distance to the platform, reflected by the lower significance levels of group differences. Animals perform different types of swimming paths, some being more focused and goal-directed (allocentric) than others (Dalm et al., Behav Res Methods Instrum Comput, 2010; Vorhees et al. Nat Protocols, 2006; Gehring et al. Sci Rep, 2015; Illouz et al. Brain Behav Immun, 2016). Some of the swimming path types might be used as strategies independent of the visual cues and are not related to hippocampal circuitry (Rogers et al., Neurobiol Learning Memory, 2017) and, therefore, do not well reflect memory assessment. On the other hand, an injury-related impairment of precision rather than spatial orientation (Kolarik et al. Neuropsychologia, 2016) would result in the animal scanning the platform quadrant for an extended amount of time before eventually finding the platform, resulting in target quadrant preference but a long distance to platform. Our collaborators for the Morris water maze (see acknowledgments) including one of the authors of this study often saw differences in learning, but not in memory (or vice versa) using different animal models.

A section discussing this issue was added in the manuscript (p. 16, l. 542-548).

Round 2

Reviewer 1 Report

The authors satisfied the reviewer's criticisms; the revised version of the manuscript reinforces the value of the results and enhances the authors' conclusions.

Author Response

Thank you again for taking your time and for accepting the changes made in the revised version of the manuscript.